# Variation of Deoxynivalenol Levels in Corn and Its Products Available in Retail Markets of Punjab, Pakistan, and Estimation of Risk Assessment

**DOI:** 10.3390/toxins13050296

**Published:** 2021-04-22

**Authors:** Shahzad Zafar Iqbal, Ahmad Faizal Abdull Razis, Sunusi Usman, Nada Basheir Ali, Muhammad Rafique Asi

**Affiliations:** 1Department of Applied Chemistry, Government College University Faisalabad, Faisalabad 38000, Pakistan; 2Department of Food Science, Faculty of Food Science and Technology, Universiti Putra Malaysia (UPM), 43400 UPM Serdang, Selangor, Malaysia; nada44basher@gmail.com; 3Natural Medicines and Products Research Laboratory, Institute of Bioscience, Universiti Putra Malaysia (UPM), 43400 UPM Serdang, Selangor, Malaysia; usunusi.bch@buk.edu.ng; 4Food Toxicology Lab, Nuclear Institute for Agriculture and Biology (NIAB), Faisalabad 38000, Pakistan; asimuhammad@yahoo.co.uk

**Keywords:** DON, corn, corn products, exposure, risk assessment

## Abstract

This study investigated the natural incidence of deoxynivalenol (DON) in corn and corn products from corn-producing districts of Punjab, Pakistan. The analysis was carried out using high performance liquid chromatography (HPLC) with UV detector and immunoaffinity cleanup columns. The detection limit (LOD) and limit of quantification were 25 and 50 µg/kg, respectively. A total of 1220 samples of corn and corn products were analyzed to detect DON, and 539 (44.2%) samples were observed to be contaminated with DON (*n* ≥ LOD). Furthermore, 92 (7.5%) samples of corn and corn products had DON levels that were higher than the proposed limits of the EU. The data are significantly different from a normal distribution of DON in samples of corn and corn products from different locations (*p* < 0.05) for Shapiro–Wilk and Kolmogorov–Smirnov values. However, a significant difference in DON levels was found between corn and corn-derived products (*p* ≤ 0.05). The lowest and highest exposures, and hazard quotient (HQ) values of 0.92 and 9.68 µg/kg bw/day, were documented in corn flour samples.

## 1. Introduction

Corn (*Zea mays*) is the third most produced crop in Pakistan after wheat and rice. Its share of the country’s total cereal cultivation area is 8.5%; it is responsible for 2.2% of value added in the agriculture sector and 0.4% of gross domestic product (GDP). Corn plays a vital role in the country’s economy through its multiple domestic, commercial and industrial uses. In 2017–2018, corn production was estimated at 5.8 million tons, 3% higher than the previous year, reflecting its in-country demand. Hybrid corn is cultivated in about 40% of the corn-growing area, and its share in total corn production is around 70%. About 65% of corn is utilized in the poultry industry, 10% is used in dairy feed, 15% is used in wet milling and the remainder is used for human consumption and seeds [1]. Corn is a cereal that is susceptible to attacks by fungi and the subsequent production of mycotoxins (e.g., deoxynivalenol (DON)) [2,3].

Mycotoxins are recognized as secondary metabolites and are produced by fungi, mainly *Aspergillus*, *Penicillium*, *Fusarium*, *Alternaria* and *Cladosporium* [4,5,6,7]. *Fusarium* is widely dispersed in nature and causes spoilage or deterioration of food and feeds [8]. DON, also identified as vomitoxin, is a secondary fungal metabolite to the class of trichothecenes and is produced by the genus *Fusarium*, especially *F. graminearum* [9,10,11,12,13]. Previous studies have documented the toxic effects of trichothecenes, such as feed refusal, hemorrhage, vomiting, diarrhea, anemia and immunosuppression [14,15,16].

In a previous study by Raza et al. [17], 63 of 100 samples from rural, semirural and urban areas in the central cities of Punjab, Pakistan, were found to be contaminated with DON. The results indicated that 63% of corn samples were contaminated with DON, and from these samples, almost 66% had levels higher than 1250 μg/kg. Furthermore, 49.2% of samples had levels higher than 1401 μg/kg. The results indicated that the levels were higher in samples from rural areas, both in corn (1512 μg/kg) and wheat (1585 μg/kg). Similarly, the effect of seasonal variation on the levels of DON in wheat and corn samples was investigated in Punjab, Pakistan [18]. The results showed that 87 (61.2%) samples of corn and corn products from summer and 57 samples (44.5%) from winter were contaminated, with concentrations ranging from 50 to 2967 µg/kg and 50 to 2490 µg/kg, respectively.

Different countries and organizations have established maximum limits for DON in raw and processed cereal products. The Codex Alimentarius Commission has established a permissible limit of 2000 µg/kg in raw barley, wheat and corn. Similarly, the EU has also established a limit of 2000 µg/kg in unprocessed wheat and oats. However, a permissible legal limit of 1000 µg/kg exists for raw or unprocessed cereals, 750 µg/kg for foodstuffs intended for consumers and dry pasta and 200 µg/kg for snacks and breakfast cereals for children [19]. The proposed daily intake limit for DON is 1 µg/kg bw.

Pakistan is located in the tropics, and therefore, the environmental conditions encourage fungal growth and production [5,20]. Although corn is a major cash crop, previous studies have shown that this substrate is more vulnerable to fungi attack and contamination. However, there is not enough information available on the prevalence of DON in corn-derived foods in Pakistan. Consequently, the existing study was designed to examine the incidence of levels of DON in corn and corn-derived products and compare these levels with EU-recommended regulations. The assessment of levels of DON in corn and corn products will also help disseminate this information to farmers, traders and other stakeholders in Pakistan.

## 2. Results

### 2.1. Method Validation

The recovery analysis of DON in corn and corn products is presented in Table 1. The recoveries were in the range of 81.3% to 91.0%, and the relative standard deviation varied from 11% to 28%. The detection limit (LOD) and limit of quantification (LOQ) were 25 and 50 µg/kg, respectively. The straight-line equation was constructed, and the coefficient of determination (0.9992) was obtained, as shown in Appendix A.

### 2.2. Incidence of DON in Corn and Corn Products

The occurrence of DON in 1220 samples of corn and corn products (449 samples from Gujranwala, 378 from Hafizabad and 393 from Sheikhupura) is presented in Table 2 and Table 3. In total, 539 samples of corn and corn products (44.2%) were found to be positive (*n* ≥ LOD) with DON. Of the 449 samples of corn and corn products from Gujranwala, 197 (43.9%) were found to be positive with DON (Table 2).

The maximum average of 1256.2 ± 124.5 µg/kg of DON was found in corn type 1 samples and varied from 25 to 5687.5 µg/kg. Table 4 shows that 166 (43.9%) of the 378 samples of corn and corn products from Hafizabad were positive, with the maximum average (1489.4 ± 190.8 µg/kg) in corn type 4 samples ranging from 25 to 5540.5 µg/kg. Furthermore, 176 (44.8%) of the 393 samples of corn and corn products from Sheikhupura were found to be contaminated with DON, and the maximum average (1489.4 ± 180.5 µg/kg) was found in corn type 4 samples, with concentration ranging from 25 to 5540.5 µg/kg. About 92 (7.5%) samples of corn and corn products were found to be contaminated with DON at levels higher than the EU’s proposed limits [19], as represented in Figure 1.

The data are significantly different from a normal distribution of DON in different types of corn and corn products from different locations, considering the high values of skewness and kurtosis and *p* ≤ 0.00 for the Shapiro–Wilk and Kolmogorov–Smirnov values. The ANOVA of DON levels in different corn and corn products was statistically significant (*p* ≤ 0.000). The amounts of DON in corn and corn products from different locations were nonsignificant (*p* ≥ 0.05), as shown in Table 5. Furthermore, Table 6 shows the least significant difference (LSD) of the different types of corn and corn products.

### 2.3. Estimation of Exposure Assessment of DON in Corn Flour

The assessment of exposure of DON levels in corn flour from different locations is presented in Table 7. The highest DON exposure was 9.68 µg/kg bw/day, and the lowest exposure was 0.92 µg/kg bw/day in corn flour samples from the Gujranwala district.

## 3. Discussion

### 3.1. Method Validation

The results showed that the analytical parameters such as accuracy and precision of DON in corn and corn products were within the European Commission’s recommended guidelines [21,22]. According to the guidelines, the recoveries of DON should be within the range of 60% to 110%, and the repeatability and reproducibility should be ≤20% and ≤40%, respectively. Furthermore, for DON values >500 µg/kg, the recommendations are 70% to 120% recoveries, and repeatability and reproducibility must be less than 20% and 40%, respectively [22,23]. The determination coefficient was ≥0.999, and LOD and LOQ were 25 and 50 µg/kg, respectively. The coefficient of determination was quite similar to the value from our previous study by Iqbal et al. [18]. However, the LOD and LOQ are much lower in the current study. Ok et al. [23] documented a coefficient of determination of ≥0.999 and found LOD and LOQ values in ranges of 6.4 to 10.0 µg/kg and 21.3 to 33.5 µg/kg, respectively. Yang et al. [24] demonstrated the LOD of 12.2 µg/kg for DON in corn samples using SPE cleanup. In another study, relatively high LOD and LOQ values for DON—i.e., 30 and 40 µg/kg—were documented in milling fractions of wheat using multifunctional column cleanup [25]. Golge and Kabak [26] determined LOQ values of 46.90 to 72.30 µg/kg of DON in various cereal products, higher than the LOQ of the present study.

### 3.2. Incidence of DON in Corn and Corn Products

The results have shown comparatively high levels of DON in corn and corn products from three districts, i.e., Gujranwala, Hafizabad and Sheikhupura in Punjab, Pakistan. In our previous study [18], we investigated the seasonal variation of DON levels in corn and corn products and documented that samples from summer have a higher incidence—i.e., 87 (61.2%)—than the samples from winter (44.5%). The maximum average observed was 1434.8 ± 25.5 µg/kg in corn flour samples from summer, and the lowest mean level was 620.8 ±17.8 µg/kg in corn bread from winter. A high incidence level of DON compared to the present results was observed in Spain, where almost 68% of commercial corn-based food samples were found to be positive, with levels ranging from 29 to 195 µg/kg [27]. In Turkey, Golge and Kabak [26] assessed 13 wheat samples (58 to 1092 µg/kg), 3 barley samples (138 to 973 µg/kg), 7 paddy rice samples (136 to 256 µg/kg), 3 wheat flour samples (92 to 151 µg/kg), 2 biscuit samples (31.2 to 71.3 µg/kg) and 1 pasta sample (59.3 µg/kg). However, two corn samples were found to be contaminated with DON (313 to 331 µg/kg). All the samples contained levels below the recommended EU limits.

A high amount of DON was documented in wheat and cereal samples from Austria, Germany, Slovakia and Australia [28] (42 to 4130 µg/kg with an average amount of 977 ± 1000 µg/kg). In Cameroon, Njobeh et al. [29] documented that 65% of 82 samples of dried food commodities contained DON levels ranging from 13 to 273 µg/kg. The occurrence and maximum amount of DON found in corn by Paladin et al. [30] in Croatia were higher than the levels found in the current study. They reported that 85% of samples were positive, with the highest concentration being 17,920 µg/kg. In Portugal, Marques et al. [31] analyzed 307 samples of plant crops, with a mean DON concentration of 70 μg/kg and a maximum amount of 17,900 μg/kg. Vidal et al. [32] from Spain observed DON in 42% of bran samples, with the highest amount being 6178 μg/kg, and 13 samples (19%) exceeded the EU’s allowable limit. In Serbia, Jajić et al. [33] documented that 32% of corn samples were contaminated; in Hungary, 86% of samples were contaminated with DON [34].

However, earlier studies have documented low levels of DON compared to those in the present survey. Escobar et al. [3] from Spain analyzed 25 samples of corn oil with mean levels of DON at 31 μg/kg. Similarly, Giménez et al. [35], also from Spain, found 10 of 25 samples of wheat germ oil with mean levels of DON at 41 μg/kg.

It is worth mentioning that regional variation was found in DON levels. Hot weather conditions promote the formation of fusarium-producing fungi and the production of mycotoxins [36]. The elevated prevalence of DON in corn and corn products in summer compared to winter was confirmed in our previous study [17]. Fernandez et al. [37] from the United States documented DON levels in cereal samples that were lower than the levels reported by Ngoko et al. [38] in corn from Cameroon. Temperate conditions favor the production of DON [39]. Furthermore, high levels of humidity combined with high temperature could enhance the growth rate of fungi belonging to the genus *Fusarium* [40]. Variation in sampling size is another factor that should be considered a significant source of variation in mycotoxins [41]. The elevated amount of DON in corn and corn products could be explained by farmers in Pakistan still using traditional methods in rural areas with cheap varieties of corn and not using crop rotation or a no-till method. Furthermore, the storage of corn in mud bins could enhance the contamination of cereal crops [42].

### 3.3. Exposure Assessment of DON in Corn Flour

In our previous study, by Iqbal et al. [18], the maximum DON exposure observed in wheat flour in summer was 8.8 µg/kg bw/day. According to the WHO [43], if ADD is greater than the provisional maximum tolerable daily intake (PMTDI), then the potential for health risk exists. The individual or population may suffer a health risk if the hazard quotient (HQ) > 1. The health risks are limited if individuals or a population are exposed to a level of DON less than 1, i.e., if HQ < 1. An exposure of 1.052 µg/kg bw/day of DON was estimated in bread and toasts [44], comparable to the exposure of DON found in the current research. An exposure of 0.027 to 0.038 µg/kg bw/day was observed by Cano-Sancho et al. [45]. The current study’s finding of a high HQ level of 1.11 in the Hafizabad and Sheikhupura districts could indicate serious risk factors for the local populations.

## 4. Conclusions

Evaluating the findings of this research and comparing them with the EU’s limit for mycotoxins reflects that the prevalence and amounts of DON were comparatively high. About 7.5% of samples of corn and corn products were contaminated with DON at levels greater than the EU’s suggested limits. The exposure and HQ amount of DON in corn and corn products were also high. Adopting good storage practices would reduce the level of DON in cereals because generally, in Pakistan, cereal crops are stored and transported in jute bags, which could absorb moisture from the environment or surface where the crop is stored. The recommendation could be to store these crops in polyethylene bags during transport and in the storage area. Furthermore, moisture, humidity and temperature should be controlled during the storage period, and farmers should use resistant crop varieties. In vivo study of DON levels in blood and urine from local consumers should be used to assess exposure and implement regulations.

## 5. Materials and Methods

### 5.1. Sampling

The 1220 samples of corn and corn products (corn flour, sweet corn, corn bread, corn oil and popcorn) from three Punjab districts (*n* = 449 Gujranwala, *n* = 378 Hafizabad, *n* = 393 Sheikhupura) were collected from June 2018 to January 2019. A simple random methodology (each portion or lot has an equal chance to be included) was used for collecting samples of corn and corn products from farmers, markets and superstores. The gross samples were taken by hand and then homogenized and properly labeled. These three districts are well-known producers of corn in Punjab, Pakistan. Due to the high variability of fungi and mycotoxins in kernel samples, each sample size was not less than 2 kg. Then, the kernel samples were mixed and ground in adequate particle size (Retsch, ZM 200, Haan, Düsseldorf, Germany). The lab sample portion was stored in plastic bags and kept in a freezer at −20 °C.

### 5.2. Chemicals and Reagents

A pure standard of DON (100 mg/mL in ACN) from Sigma-Aldrich (St. Louis, MO, USA) was available in the lab (a food safety lab). High-grade purity (≥99%) solvents of acetonitrile, methanol and polyethylene glycol 8000 (PEG) were acquired from Sigma-Aldrich (Sigma-Aldrich, Lyon, France). The method’s linearity was assessed with seven-point concentrations (100, 200, 800, 1600, 3200, 6400 and 9000 µg/kg) of DON and stored in capped vials at a temperature of −20 °C. All other chemicals used were of analytical grade and available in the lab.

### 5.3. Extraction and HPLC Parameters

The extraction of DON in corn and corn products was carried out using our previously validated method [17]. Briefly, the sample (5 g) and PEG (1 g) were mixed in 20 mL of ultrapure water, homogenized in 50 mL Teflon tubes and centrifuged at room temperature for 1 min at 6500 rpm. The extraction of corn bread was carried out by adding 200 mL water to dry bread and then homogenizing for 5 min at room temperature at 8000 rpm, as discussed above. After centrifugation, the mixture was filtered and the filtrate (5 mL) was passed through to the immunoaffinity column (IAC-NIV WB, columns) (VICAM, Watertown, MA, USA). The column was washed twice with 10 mL purified water, and the DON was eluted using 1 mL of pure methanol from the IAC column. Then, 0.5 mL of pure water was added to 0.5 mL of eluate and subjected to HPLC analysis, after passing through 0.22 µm nylon syringe filters. The study used a Shimadzu (Kyoto, Japan) HPLC system with a C18 Supelco column (250 × 4.6 × 5 mm) (Discovery HS, Bellefonte, PA, USA). The UV detector (RF-530) was set at a detection wavelength of 218 nm. The mobile phase was 30% methanol and 70% water, with a flow rate of 1.2 mL/min.

### 5.4. Exposure Assessment

According to the literature, the level of DON found in food in the present study is considered reasonable. Further, it was believed that exposure to DON was only through corn flour, this being the main ingredient in the Pakistani diet. The mean and highest levels of DON in corn flour were used to assess exposure, and an adult body weight is regarded as 65 kg bw. The per capita consumption of corn flour was 0.07 kg [1]. The exposure was assessed using the formula of Equation (1):(1)$${\mathrm{ADD}}_{\mathrm{DON}}\;=\frac{C\;DON\;\times \;IR}{bw}$$
where ADD_DON_ is the average daily dose of DON, *C_DON_* is the concentration of *DON* in corn flour, *IR* is the intake rate of corn flour and *bw* is body weight.

The hazard quotient (HQ) values for the mean and highest levels of DON in corn flour were also estimated and equaled to exposure assessment studies. The provisional maximum tolerable daily intake (*PMTDI*) was 1 µg/kg bw for DON [20]. The HQ was determined following the formula shown in Equation (2):(2)$$\mathrm{HQ}\;=\;\frac{\mathrm{ADD}\;\mathrm{DON}\;}{PMTDI}$$

### 5.5. Method Validation

All the method validation parameters were performed for the assessment of DON in corn and corn products. The recovery analysis was performed. The fortified levels (100, 150, 300, 400, 800, 3000 µg/kg) of DON were added in uncontaminated samples of corn and corn product samples. The LOD and LOQ were assessed as 3:1 and 10:1 signal-to-noise ratios, respectively.

### 5.6. Statistical Analysis

The results of DON levels in corn and corn products were subjected to statistical analysis using SPSS (version 26 for Windows, SPSS Inc., Chicago, USA). The DON levels in different corn samples and from different locations were checked for normal distribution (Shapiro–Wilks test). ANOVA was applied to investigate the difference of means (DON levels) between different types of corn and corn products and from different locations. The least significant difference (LSD) was used to investigate differences within each type or each location. A probability value of 0.05 was used to determine the statistical significance.

## Figures and Tables

**Figure 1 toxins-13-00296-f001:**
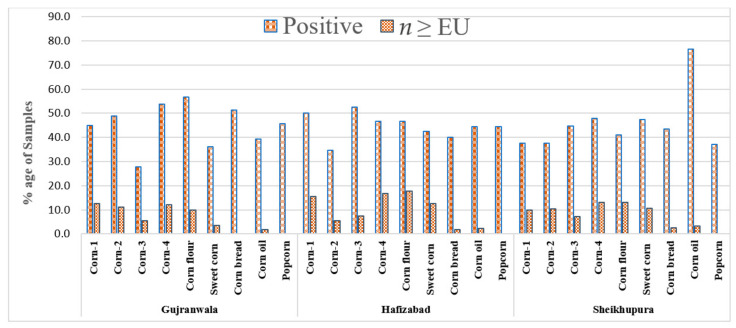
Percentage of samples of corn and corn products having levels of deoxynivalenol (DON) higher than the EU’s recommended limits (1000 µg/kg for unprocessed corn and 750 µg/kg for foodstuffs).

**Table 1 toxins-13-00296-t001:** Recovery percentage of deoxynivalenol (DON) in corn and corn products.

DON Level	Corn (Mix)				Corn Flour				Corn Bread				Sweet Corn			
	Recovery	RSD	Precision (%)		Recovery	RSD	Precision (%)		Recovery	RSD	Precision (%)		Recovery	RSD	Precision (%)	
µg/kg	(%)	(%)	Rep ^a^	Reprod ^b^	(%)	(%)	Rep ^a^	Reprod ^b^	(%)	(%)	Rep ^a^	Reprod ^b^	(%)	(%)	Rep ^a^	Reprod ^b^
100	84.5	16	11	21	81.5	11	12	19	89.5	14	9	20	85.2	16	10	21
150	89.5	11	18	19	86.5	14	09	15	83.5	18	15	26	89.5	17	14	17
300	88.0	22	15	22	87.4	17	16	22	90.5	23	10	18	89.5	15	9	21
400	91.0	13	18	24	90.9	12	14	20	89.5	22	12	22	88.4	19	17	26
800	88.8	23	10	22	86.1	16	18	26	88.4	21	15	20	85.2	21	15	24
3000	85.0	21	18	28	84.0	20	20	27	85.5	20	18	27	81.3	28	18	28

RSD = relative standard deviation, LOD = 25 µg/kg, LOQ = 50 µg/kg, a = repeatability, b = reproducibility.

**Table 2 toxins-13-00296-t002:** Occurrence of DON in corn and corn products from Gujranwala, Punjab, Pakistan.

Sample Type	Samples *N*	Positive *N* (%)	Mean (µg/kg) ± S.D.	Range (µg/kg)
Corn 1	40	18 (45.0)	1256.2 ± 124.5	25–5687.5
Corn 2	45	22 (48.9)	1126.2 ± 90.8	25–5568.6
Corn 3	54	15 (27.8)	1049.7 ± 110.5	25–5678.5
Corn 4	41	22 (53.7)	909.8 ± 120.5	25–4550.5
Corn flour	60	34 (57.6)	854.7 ± 40.7	25–8990.5
Sweet corn	83	30 (36.1)	556.5 ± 80.6	25–4550.5
Corn bread	35	18 (51.4)	86.6 ± 14.5	25–450.5
Corn oil	56	22 (39.3)	88.8 ± 15.5	25–980.5
Popcorn	35	16 (45.7	41.4 ± 10.5	25–95.5
Total	449	197 (43.9)		25–8990.5

Corn 1 = corn variety; Corn 2 = corn variety; Corn 3 = corn variety; Corn 4 = corn variety, *N* (%) = *n* (percentage of samples), LOD = 25 µg/kg.

**Table 3 toxins-13-00296-t003:** Occurrence of DON in corn and corn products from Sheikhupura, Punjab, Pakistan.

Sample Type	Samples *N*	Positive *N* (%)	Mean (µg/kg) ± S.D.	Range (µg/kg)
Corn 1	48	15 (37.5	1187.3 ± 230	25–6660.5
Corn 2	48	18 (37.5)	1081.6 ± 145.5	25–6590.6
Corn 3	56	25 (44.6)	808.2 ± 60.5	25–5640.5
Corn 4	46	22 (47.8)	1489.4 ± 180.5	25–5540.5
Corn flour	61	25 (41.0)	1030.8 ± 170.5	25–7860.5
Sweet corn	38	18 (47.4)	471.1 ± 30.5	25–2345.5
Corn bread	39	17 (43.6)	97.3 ± 15.5	25–500.5
Corn oil	30	23 (76.7)	88.3 ± 9.5	25–540.5
Popcorn	35	13 (37.1)	42.1 ± 4.5	25–90.5
Total	393	176 (44.8)		25–7860

Corn 1 = corn variety; Corn 2 = corn variety; Corn 3 = corn variety; Corn 4 = corn variety, *N* (%) = *n* (percentage of samples), LOD = 25 µg/kg.

**Table 4 toxins-13-00296-t004:** Occurrence of DON in corn and corn products from Hafizabad, Punjab, Pakistan.

Sample Type	Samples *N*	Positive *N* (%)	Mean (µg/kg) ± S.D.	Range (µg/kg)
Corn 1	32	16 (50.0)	1187.3 ± 210.5	25–6660.5
Corn 2	55	19 (34.5)	1081.6 ± 140.3	25–6590.6
Corn 3	40	21 (52.5)	808.2 ± 70.5	25–5640.5
Corn 4	30	14 (46.7)	1489.4 ± 190.8	25–5540.5
Corn flour	45	21 (46.7)	1030.8 ± 105.4	25–7860.5
Sweet corn	40	17 (42.5)	471.1 ± 20.5	25–2345.5
Corn bread	55	22 (40.0)	97.3 ± 15.5	25–500.5
Corn oil	45	20 (44.4)	88.3 ± 9.5	25–540.5
Popcorn	36	16 (44.4)	42.2 ± 4.5	25–80.5
Total	378	166 (43.9)		25–7860.5

Corn 1 = corn variety; Corn 2 = corn variety; Corn 3 = corn variety; Corn 4 = corn variety, *N* (%) = *n* (percentage of samples), LOD = 25 µg/kg.

**Table 5 toxins-13-00296-t005:** ANOVA of DON levels in corn and corn products from various locations.

		Sum of Squares	df	Mean Square	F	Sig
DON Level	Between groups	96,192,972.7	8	12,024,121.6	8.132	0.000
	Within groups	783,676,623.6	530	1,478,635.1		
	Total	879,869,596.4	538			
Location	Between groups	4.737	8			
	Within groups	367.4	530	0.592	0.854	0.555
	Total	372.2	538	0.693		

**Table 6 toxins-13-00296-t006:** Least significant difference (LSD) analysis of different types of corn and corn products within groups of samples.

		Corn Products	Mean Difference	Std. Error	Significance
**Don Level**	**Corn 1**	Sweet corn	629.92	230.05	0.006
		Corn bread	1066.63	236.89	0.000
		Corn oil	1073.58	230.05	0.000
		Popcorn	1130.06	251.06	0.000
	**Corn 2**	Sweet corn	542.88	218.65	0.013
		Corn bread	979.59	225.83	0.000
		Corn oil	986.53	218.65	0.000
		Popcorn	1043.03	240.66	0.000
	**Corn 3**	Corn bread	842.21	224.01	0.000
		Corn oil	849.16	216.76	0.000
		Popcorn	9.05.65	238.95	0.000
	**Corn 4**	Sweet corn	483.11	219.64	0.028
		Corn bread	919.82	226.79	0.000
		Corn oil	926.77	219.64	0.000
		Popcorn	983.26	241.56	0.000
	**Corn flour**	Corn bread	772.55	210.76	0.000
		Corn oil	779.50	203.05	0.000
		Corn popcorn	8.35.99	226.58	0.000
	**Sweet corn**	Corn type 1	−629.91	230.05	0.006
		Corn type 2	−542.88	218.65	0.013
		Corn type 4	−483.11	219.64	0.028
		Corn bread	436.70	220.65	0.048
		Corn oil	443.65	213.29	0.038
		Popcorn	500.15	235.81	0.034
	**Corn bread**	Corn type 1	−1066.62	236.89	0.000
		Corn type 2	−979.59	225.83	0.000
		Corn type 3	−842.21	224.01	0.000
		Corn type 4	−919.82	226.79	0.000
		Corn flour	−772.55	210.76	0.000
		Sweet corn	−436.70	220.65	0.048
	**Corn oil**	Corn type 1	−1073.57	230.05	0.000
		Corn type 2	−986.53	218.65	0.000
		Corn type 3	−849.15	216.76	0.000
		Corn type 4	−926.73	219.64	0.000
		Corn flour	−779.50	203.05	0.000
		Sweet corn	−443.65	213.29	0.038
	**Popcorn**	Corn type 1	−1130.06	251.06	0.000
		Corn type 2	−1043.03	240.66	0.000
		Corn type 3	−905.65	238.95	0.000
		Corn type 4	−983.26	241.56	0.000
		Corn flour	−835.99	226.58	0.000
		Sweet corn	−500.15	235.81	0.034

d.f. = degree of freedom; F = F test; significance = *p* value.

**Table 7 toxins-13-00296-t007:** Exposure assessment of DON in corn flour from corn-producing cities of Punjab, Pakistan.

	Consumption (kg)	Gujranwala				Hafizabad				Sheikhupura			
		DON Levels		Exposure		DON Levels		Exposure		DON Levels		Exposure	
		Mean µg/kg	Highest Level µg/kg	Mean µg/kg bw/day	Highest µg/kg bw/day	Mean µg/kg	Highest Level µg/kg	Mean µg/kg bw/day	Highest µg/kg bw/day	Mean µg/kg	Highest Level µg/kg	Mean µg/kg bw/day	Highest µg/kg bw/day
Corn flour	0.07	854.7	8990.5	0.92	9.68	1030.8	7860.5	1.11	8.47	1030.8	7860.5	1.11	8.47
HQ ^1^				0.92				1.11				1.11	
HQ ^2^					9.68				8.47				8.47

HQ ^1^ = mean level of DON, HQ ^2^ = highest level of DON.

## Data Availability

The data will be available upon request.

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
