# Peer review of "Variation of Deoxynivalenol Levels in Corn and Its Products Available in Retail Markets of Punjab, Pakistan, and Estimation of Risk Assessment"

_toxins, 2021, doi:10.3390/toxins13050296_

Round 1

Reviewer 1 Report

Although this paper proposes to inform about the surveillance for contamination levels of DON in maize and Pakistani corn-retail food, I did not find enough evidence in the paper to justify its publication in toxins.

1-There is not an original study. Similar ones have been published and some of these works were not acknowledgment in the present article.

2- The authors used HPLC-UV methodology to detect and determine DON in the samples. Although the HPLC-UV is widely used, references of this protocol are needed. Moreover, the identity of DON parameter should be confirmed by LC-MS, GC-MS or LC-MS/MS.  DON is a mycotoxin that is not strongly absorb UV.

3-Which Vicam- IAF column did you use? Did you use DON test HPLC column? The lower detected level associated to the column should be specified.

4-Validation:

  1. The fortification procedure should include samples spiked at lower concentrations. For example, final concentrations such as 30- 50 µg/Kg DON should to be included. The lower levels are the more critical and many samples were reported with levels <100 µg/Kg DON.
  2. An interlaboratory validation is suggested.
  3. Linearity and range should be visualized (it could be shown as supplementary graphic). Also, the matrix effect should to be considered.

5- Only one mycotoxin level was analyzed. Humans and animals are typically exposed to mycotoxin mixtures which can induce combined adverse health effects. The European Commission recognizes the need for simultaneous analysis of Fusarium toxins.

6-Sampling:

  1. More data about corn samples are needed. Kind of corn, storage corn, etc.
  2. Only one year? Which year were the samples collected?

7-Which is the risks associated with the intake of cornflour samples in children?

8-The introduction should have all the necessary information to understand the study. Previous related studies (especially the studies conducted in Pakistan), analytical methods, regulatory levels and standards impact for processed an unprocessed food, PMTDI, etc. 

Line 17- This line doesn’t belong to this study.

Author Response

The Editor

Toxins (ISSN 2072-6651)

Dated: 9thMarch 2021

Please find enclosed our manuscript toxins-1111627 R1 titled “Variation of Deoxynivalenol Levels in Corn and Its Products Available in Retail Markets of Punjab, Pakistan and Estimation of Risk Assessment. The manuscript is revised extensively as suggested by Reviewer 1. First, we are grateful for the time and critical comments raised by our respected reviewer no 1. It is an excellent critical review, constructive comments as well as important suggestion for improvements in our future studies. The response for each comment is given and addition in manuscript is highlighted as yellow font.

Reviewer No 1:

Although this paper proposes to inform about the surveillance for contamination levels of DON in maize and Pakistani corn-retail food, I did not find enough evidence in the paper to justify its publication in toxins.

Response: Thanks so much, we really honor for your justification and your point of view.

Comment: 1-There is not an original study. Similar ones have been published and some of these works were not acknowledgment in the present article.

Response: The matter of fact is that we have done analysis under our common lab, Food Toxicology Lab, NIAB, Faisalabad, Pakistan, and Dr. Muhammad Rafique Asi was our mentor. He was died in this December 2021 due to covid-19 complication, and its really shocking for me especially. We have been working since 2004 and we have almost 55-60 publications in jointly, he has established lab for mycotoxins analysis and we have been contributing equally and jointly. Our lab recent publications on DON from Pakistan are

  1. Raza, H.M.F.; Asi, M.R.; Maqbool, U. Assessment of deoxynivalenol (don) mycotoxin in corn and wheat grains consumed in central Punjab, Pakistan. J. Bot. 2020,  52(6), 2205-2210.
  2. Iqbal, S.Z.; Usman, S.; Razis, A.F.A.; Ali, N.B.; Saif, T.; Asi, M.R. Assessment of deoxynivalenol in wheat, corn and products and estimation of dietary intake. Int. J. Environ. Res. Public Health, 2020, 17 (15), 5602.

We have greatly used UV detector and its results were great as mentioned in study no 1. Furthermore, I would really like to dedicate this work to him, he was so good both in method validation and method development and his lab got ISO certification as well.

Comment 2- The authors used HPLC-UV methodology to detect and determine DON in the samples. Although the HPLC-UV is widely used, references of this protocol are needed. Moreover, the identity of DON parameter should be confirmed by LC-MS, GC-MS or LC-MS/MS.  DON is a mycotoxin that is not strongly absorb UV.

Response: Its very good comment and important as well. The matter of fact is that in Pakistan, the HPLC analysis is quite easily you can carry on, but LC-MS/MS are difficult to take your turn and expensive too. The present Government is giving funds to Universities and hope we will able to utilize LC-MS/MS in our future studies too.  The references of HPLC-UV detector are provided in earlier comments, our lab latest studies.

3-Which Vicam- IAF column did you use? Did you use DON test HPLC column? The lower detected level associated to the column should be specified.

Response: The DON column (NIV WB, columns) were used and C18 column was used as reverse phase HPLC for the analysis of DON.

4-Validation:

  1. The fortification procedure should include samples spiked at lower concentrations. For example, final concentrations such as 30- 50 µg/Kg DON should to be included. The lower levels are the more critical and many samples were reported with levels <100 µg/Kg DON.

Response: We have used the lowest value of calibration curve of 50 µg/L, but this value gives very high standard deviation, so we intentionally did not mention this value in Material and method section. However, during calculation we use the lower 50 µg/L (quantification limit).   

  1. An interlaboratory validation is suggested.

Response: Its very good comment to estimate and visualize our inaccuracies but every lab has not mycotoxins setup, so very difficult, if not impossible.

  1. Linearity and range should be visualized (it could be shown as supplementary graphic). Also, the matrix effect should to be considered.

Response: The calibration curve has been presented in supplementary graphics, in our first study we have analyzed the matrix effect and used various kind of separating columns too, but we found Vicam columns give best recovery.

5- Only one mycotoxin level was analyzed. Humans and animals are typically exposed to mycotoxin mixtures which can induce combined adverse health effects. The European Commission recognizes the need for simultaneous analysis of Fusarium toxins.

Response: Its very good and critical point, actually this project is based on DON detection in cereal samples only, and our collaborative Dr. Asi was well versed on detection and validation of different types of mycotoxins in cereals and other food stuff. But we will have plans to do study on simultaneous analysis of DON, ZEN, FB1 and FB2 analysis. Just shortcoming of funding at present movement. 

6-Sampling:

  1. More data about corn samples are needed. Kind of corn, storage corn, etc.
  2. Only one year? Which year were the samples collected?

Response: The samples were collected during June 2018 to January 2019 and not stored long before they were analyzed because we have to finish our ongoing project. The detail has also been included in manuscript too.

7-Which is the risks associated with the intake of cornflour samples in children?

Response: The most intensive use of corn flour is corn flour bread which mostly consumed by adult person in Pakistan. Therefore, we focus exposure assessment to adults only.

8-The introduction should have all the necessary information to understand the study. Previous related studies (especially the studies conducted in Pakistan), analytical methods, regulatory levels and standards impact for processed an unprocessed food, PMTDI, etc. 

Response: Thanks so much for your suggestion, we have revised introduction section, according to your suggestions and highlighted as yellow font as “

 In previous study, high levels of DON were documented by Raza et al. [16], from ru-ral, semi-rural and urban areas from central cities of Punjab, Pakistan. The results had in-dicated that 63% samples of corn were contaminated with DON, and from these samples almost 66% samples have levels higher than 1250 μg/kg. Furthermore, 49.2% samples have levels highly than 1401 μg/kg. The results have indicated that the levels were higher in samples from rural areas samples for both in corn (1512 μg/kg) and wheat (1585 μg/kg) grains. Similarly, the effect of seasonal variation on the levels of DON in wheat and corn samples were investigated from Punjab, Pakistan [17]. The results have shown that 104 (44.8%) samples of wheat and products from the summer season, and 91 (41.9%) samples from winter season were found to be contaminated with DON with levels ranged from LOD to 2145 µg/kg and LOD to 2050 µg/kg, respectively. However, 87 (61.2%) samples of corn and products from summer and 57 (44.5%) from winter season were found to be contaminated with concentration ranged from LOD to 2967 µg/kg, and LOD to 2490 µg/kg, respectively.

 The countries and organization have established maximum limits for DON in raw and processed cereal products. The Codex Alimentarius Commission has established permissible limit of 2000 µg/kg in raw barley, wheat, and corn.  Similarly, EU has also established a limit of 2000 µg/kg in unprocessed wheat, and oat. However, a permissible legal limit of 1000 µg/kg for raw or those cereals which would be unprocessed, and 750 µg/kg for foodstuffs, intended for consumers and dry pasta and 200 µg/kg for snacks and breakfast cereals used for children [ 18]. The tolerance daily intake for DON is proposed as 1 µg/kg/bw per day.  

Line 17- This line doesn’t belong to this study.

Response: Sorry, of course this line is not belonged to this study and its deleted as well.

Thanks once again, for providing in-depth of your critical comments.

Reviewer 2 Report

Thank you for the opportunity to review this article. The manuscript deals with the topic: Variation of Deoxynivalenol Levels in Corn and Its Products Available in Retail Markets of Punjab, Pakistan and Estimation of Risk Assessment.

The article is well written with exception to a few minor points.

Please in the introduction provide sufficient background and include more relevant references.

Moreover

Line 9: The sentence “of corn & products … please change the &

Line 14: highest exposure & hazard… please change the &

Line 17: A total of 2970 samples of twelve citrus fruits types were analyzed for patulin 17 (PAT) contamination…. Patulin????  citrus fruits??????

Line 18-19:  corn & products from… please change the &

Line 25: Zea mays…please write in the same font size.

Line 26-27, 29:  Change all the percent with %

Line 35-36: Mycotoxins are recognized as secondary metabolites and produced by fungi, mainly 35 Aspergillus, Penicillium, Fusarium, Alternaria, and Cladosporium [4-6]… Quite old bibliography. Please add the latest one Agriopoulou, S., Stamatelopoulou, E. Varzakas, T. (2020). Advances in occurrence, importance and mycotoxin control strategies: prevention and detoxification in foods. Foods. 9, 137.

Line 58: Write the first line of the Table 1 with the same way. Put parenthesis at the first, second and fourth RSD

Line 61: samples of corn and products…. and corn products.

Line 64: samples of corn and corn… please correct.  

Line 67: put the fullstop at the end of the phrase

Line 67: Please correct the word LOD in last line.

Line 67: Corn 1= variety of corn; corn 2= corn variety; corn 3 = variety of corn; corn 4= variety of corn…What is the difference between variety of corn and corn variety?

Line 67: Please write the column Range with the same way.

Line 80: corn and products… please correct.  

Line 83: Corn 1= variety of corn; corn 2= corn variety; corn 3 = variety of corn; corn 4= variety of corn…What is the difference between variety of corn and corn variety?

Line 96: Explain the meaning of df, F and Sig

Line 123: Golge & Kabak,…. please change the &

Line 126, 129: in corn and products… please correct.  

Line 135: Golge, & Kabak… please change the &

Line 141: Please compete the sentence.

Line 158: Could you provide some climatic data from the regions which samples were taken? Like moisture, humidity, and temperature.

Line 158: fungi'…please correct.

Line 166: fusarium…  Fusarium

Line 193: The Vivo study… In vivo

Line 198: The 1220 samples of corn …which year?

Line 267-268: Write in the same font size.

Line 292: 2012. ….without fullstop

Line 294: Int. J. Environ. Res. Public Health, 2020… 2020 bold

Author Response

The Editor

Toxins (ISSN 2072-6651)

Dated: 9th March 2021

Please find enclosed our manuscript toxins-1111627 R1 titled “Variation of Deoxynivalenol Levels in Corn and Its Products Available in Retail Markets of Punjab, Pakistan and Estimation of Risk Assessment. The manuscript is revised extensively as suggested by Reviewer no 2. First, we are grateful for the time, commitment and professional approach during the review of our manuscript. It is an excellent review, unbiased, constructive critical comments as well as important suggestion for improvements. We really appreciate the commitment of reviewer and hope a positive outcome here.

Reviewer No 2Thank you for the opportunity to review this article. The manuscript deals with the topic: Variation of Deoxynivalenol Levels in Corn and Its Products Available in Retail Markets of Punjab, Pakistan and Estimation of Risk Assessment.

The article is well written with exception to a few minor points.

Response: Thanks so much and we extensively revise the manuscript as we can, hope to hear a positive news from you.

Please in the introduction provide sufficient background and include more relevant references.

Moreover

Line 9: The sentence “of corn & products … please change the &

Response: Changed and highlighted

Line 14: highest exposure & hazard… please change the &

Response: changed

Line 17: A total of 2970 samples of twelve citrus fruits types were analyzed for patulin 17 (PAT) contamination…. Patulin????  citrus fruits??????

Response: Sorry, for confusion and this sentence is deleted. So sorry

Line 18-19:  corn & products from… please change the &

Response: changed

Line 25: Zea mays…please write in the same font size.

Response: changed

Line 26-27, 29:  Change all the percent with %

Response: Changed accordingly

Line 35-36: Mycotoxins are recognized as secondary metabolites and produced by fungi, mainly 35 AspergillusPenicilliumFusariumAlternaria, and Cladosporium [4-6]… Quite old bibliography. Please add the latest one Agriopoulou, S., Stamatelopoulou, E. Varzakas, T. Advances in occurrence, importance and mycotoxin control strategies: prevention and detoxification in foods. Foods. 2020, 9, 137.

Response: Thanks, we included the desired updated reference

Line 58: Write the first line of the Table 1 with the same way. Put parenthesis at the first, second and fourth RSD

Response: changed

Line 61: samples of corn and products…. and corn products.

Response: changed as suggested

Line 64: samples of corn and corn… please correct.  

Response: changed accordingly

Line 67: put the fullstop at the end of the phrase

Response: changed

Line 67: Please correct the word LOD in last line.

Response: changed

Line 67: Corn 1= variety of corn; corn 2= corn variety; corn 3 = variety of corn; corn 4= variety of corn…What is the difference between variety of corn and corn variety?

Response: The whole format is changed as “ Corn verity; corn 2= corn variety; corn 3 = Corn variety; corn 4= Corn variety”

Line 67: Please write the column Range with the same way.

Response: ranges were changed

Line 80: corn and products… please correct.  

Response: changed

Line 83: Corn 1= variety of corn; corn 2= corn variety; corn 3 = variety of corn; corn 4= variety of corn…What is the difference between variety of corn and corn variety?

Response: The pattern was changed as “ Corn 1= corn variety; corn 2= corn variety; corn 3 = corn variety; corn 4= corn variety,

Line 96: Explain the meaning of df, F and Sig

Response: These terms are highlighted in footer of table

Line 123: Golge & Kabak,…. please change the &

Response: Changed

Line 126, 129: in corn and products… please correct.

Response: changed accordingly

Line 135: Golge, & Kabak… please change the &

Response: changed

Line 141: Please compete the sentence.

Response: changed

Line 158: Could you provide some climatic data from the regions which samples were taken? Like moisture, humidity, and temperature.

Response: Its important comment, unfortunately, climatic data was not recorded during analysis and in coming study we are up to on moisture and humidity levels.

Line 158: fungi'…please correct.

Response: changed

Line 166: fusarium…  Fusarium

Response: corrected

Line 193: The Vivo study… In vivo

Response: The sentence is changed and revised

Line 198: The 1220 samples of corn …which year?

Response: The year is included

Line 267-268: Write in the same font size.

Response: Changed accordingly

Line 292: 2012. ….without fullstop

Response: changed

Line 294: Int. J. Environ. Res. Public Health, 2020… 2020 bold

Response: changed

We thank reviewer once again

Reviewer 3 Report

The article titled Variation of Deoxynivalenol Levels in Corn and Its Products Available in Retail Markets of Punjab, Pakistan and Estimation of Risk Assessment is an interesting study of DON concentrations in a total of 1220 samples. However, there are some points that need improvement:

- Line 17: the first sentence in Key contribution is:  A total of 2970 samples of twelve citrus fruits types were analyzed for patulin 17 (PAT) contamination? This is an error. Delete the sentence.

- Line 19 and Line 20: delete the word about.

- Line 47:  the existing study was designed to examine the incidence levels of DON in corn and corn-derived products (add »in corn«)

- Table 1: DON not Don (first column)

- Line 63: delete word about. Show the exact number in the table.

- Line 65: corn products

- Table 2: explain the type of samples Corn 1-4: my suggestion is to use a number instead of LOD

- Line 74: change the word infected

- Line 77: EU proposed limits – reference is missing, which is vital for your study and results.

- Figure 1: y-axis - % age – delate age

- Table 5: same as for Table 1

- Line 197, Subchapter Sampling: Please, describe the sampling method used to ensure the representativeness of the samples.

- Line 200: sheikhupura – the first letter must be uppercase.

- Line 210: The method's linearity was assessed with seven-point concentrations (100, 200, 800, 1600, 3200, 6400, and 9000 μg/kg). But your LOD and LOQ are 25 μg/kg in 50 μg/kg. How do you control that?

- Line 234: The per capita consumption of cornflour was 0.07 kg (1). Where in Reference 1 is this data?

- Line 243: I could not find a reference under number 18.

- Line 259: space is missing between 0.05 and was

Author Response

The Editor

Toxins (ISSN 2072-6651)

Dated: 9th March 2021

Please find enclosed our manuscript toxins-1111627 R1 titled “Variation of Deoxynivalenol Levels in Corn and Its Products Available in Retail Markets of Punjab, Pakistan and Estimation of Risk Assessment. The manuscript is revised extensively as suggested by Reviewer no 3. First, we are really grateful for the time, efforts and impartial review and comments suggested by our respected reviewer no 3. It can be included as one of the excellent reviews, in past 4-5 years period. The response is given each point and revision is done carefully. We hope the positive outcome from reviewer no 3. Thanks once again.

Reviewer No 3: The article titled Variation of Deoxynivalenol Levels in Corn and Its Products Available in Retail Markets of Punjab, Pakistan and Estimation of Risk Assessment is an interesting study of DON concentrations in a total of 1220 samples. However, there are some points that need improvement:

Response: Thanks so much for your encouragement and positive review

- Line 17: the first sentence in Key contribution is:  A total of 2970 samples of twelve citrus fruits types were analyzed for patulin 17 (PAT) contamination? This is an error. Delete the sentence.

Response: Thanks so much, really deeply sorry, and the line is deleted in revised version

- Line 19 and Line 20: delete the word about.

Response: Thanks, the word about is deleted in both lines and changed as “

Samples 539 (44.2%) of corn & products were found to be positive with DON (n ≥ LOD). Samples 92 (7.5%) of corn and corn derived products having levels of DON greater than the proposed limits of EU”

Comment:  Line 47:  the existing study was designed to examine the incidence levels of DON in corn and corn-derived products (add »in corn«)

Response: Thanks so much, the suggested changes is incorporated as “

“Consequently, the existing study was designed to examine the incidence levels of DON in corn and corn-derived products and------------

Comment : Table 1: DON not Don (first column)

Response: Thanks, again our mistake, in revised version this mistake is corrected

- Line 63: delete word about. Show the exact number in the table.

Response: The sentence is changes as

“Samples 539 (44.2%) were found to be positive------------

- Line 65: corn products

Response: changed

Comment: Table 2: explain the type of samples Corn 1-4: my suggestion is to use a number instead of LOD

Response: The verities types, occurrence of DON and their nutritional values are planning to submit in another manuscript, the values of LOD are included now.

- Line 74: change the word infected

Response: changed to “contaminated”

- Line 77: EU proposed limits – reference is missing, which is vital for your study and results.

Response: The reference is provided as “[18]

- Figure 1: y-axis - % age – delate age

Response: deleted

- Table 5: same as for Table 1

Response: changed

- Line 197, Subchapter Sampling: Please, describe the sampling method used to ensure the representativeness of the samples.

Response: The line is added as “

“A simple random methodology was used for collecting corn and corn products samples from farmers, market………

- Line 200: sheikhupura – the first letter must be uppercase.

Response: changed

- Line 210: The method's linearity was assessed with seven-point concentrations (100, 200, 800, 1600, 3200, 6400, and 9000 μg/kg). But your LOD and LOQ are 25 μg/kg in 50 μg/kg. How do you control that?

Response: We have used the lowest point of detection limit of 50 µg/L, however it generate greater levels of S.D. values therefore we intentionally not used in Material and method section. However, calculation were done using standard curve as presented in supplementary section

- Line 234: The per capita consumption of cornflour was 0.07 kg (1). Where in Reference 1 is this data?

Response: The reference is given as “

  1. USDA. Pakistan: Grain and Feed Annual. GAIN Report Number:PK1707, 2017.Assessed on 24th January 2021.

- Line 243: I could not find a reference under number 18.

Response: The reference is given as

  1. European Commission. Commission Regulation No. 1881/2006 of 19 December 2006 setting maximum levels for certain contaminants in foodstuff. Eur. J. Union 2006, 364, 5–24.

- Line 259: space is missing between 0.05 and was

Response: changed

Thanks so much once again, and we hope a positive and kind response.

Reviewer 4 Report

In experiment based on investigating the natural incidence of deoxynivalenol (DON) in corn and products from corn producing districts of Punjab, Pakistan the authors confirmed the danger of the presence of high levels of DON in these products. In my opinion this manuscript is interesting, but it need a few corrections:
Line 10 and Figure 1: Is it correct to define n> LOD and n> EU? Please check it out
Line 5 and Line 34 and Line 37: the abbreviation of the name should be explained when the word is used for the first time
Table 1: Please change Don to DON ad LOD=25 µg/kg.
There is no clear explanation regarding Repeatability and Reproducibility. In table are Reprod and Rep, but lack of these abberviations in legend
Line 73: lack of µg/kg for LOD
Line 91: p<0.000 Is it a mistake?
Line 96 I propose to write down as Table 5a. instead of Table 5. (a):
Line 99: The same suggestion - Table 5b.
Line 111: Please insert the reference number instead of (EC, 2006).
Line 141: 977+1000 ug/kg - Has standard deviation been calculated correctly?
Line 166: Please change it to Fusarium italica
Line 173-181: There are a few unit and editing errors i.e. µg/kg day b.w. instead of µg/kg b.w./day

Line 193-194: Please correct this sentence
Line 207: A pure standard of DON (100 mg/mL in ACN)… Please correct
Line 219: 5 minutes Please correct
Line 226: I suggest to (250 x 4.6 x 5 mm). Please correct 
Line 236 and line 245: I suggest working on the patterns to improve their quality.
Line 267-268; 272; 283; 285; 292; 294; 298-299; 307 - I suggest to improve the literature items according to the journal's requirements

Author Response

The Editor

Toxins (ISSN 2072-6651)

Dated: 9th March 2021

Please find enclosed our manuscript toxins-1111627 R1 titled “Variation of Deoxynivalenol Levels in Corn and Its Products Available in Retail Markets of Punjab, Pakistan and Estimation of Risk Assessment. The manuscript is revised extensively as suggested by Reviewer no 4. First, we are really grateful for the time, commitment and professional dedication of reviewer no 4. It is an excellent review no doubt about that and we have revised the whole manuscript extensively. We hope a positive and kind outcome from you. Thanks

Reviewer No 4: In experiment based on investigating the natural incidence of deoxynivalenol (DON) in corn and products from corn producing districts of Punjab, Pakistan the authors confirmed the danger of the presence of high levels of DON in these products. In my opinion this manuscript is interesting, but it need a few corrections:

Response: Thanks so much for your encouragement and evaluation. It reflects your professional approach, broad and open thinking and unbiased observation.

Line 10 and Figure 1: Is it correct to define n> LOD and n> EU? Please check it out

Response: it is confirmed n ≥ EU

Line 5 and Line 34 and Line 37: the abbreviation of the name should be explained when the word is used for the first time

Response: The mistake is corrected, thanks so much for pointing out it and improving the quality of our manuscript.

Table 1: Please change Don to DON ad LOD=25 µg/kg.

Response: Thanks, the suggested changes have been incorporated

There is no clear explanation regarding Repeatability and Reproducibility. In table are Reprod and Rep, but lack of these abberviations in legend

Response: So sorry for it, we have provided the full description as table footer and highlighted in revised manuscript as well, sorry for confusion

Comment: Line 73: lack of µg/kg for LOD

Response: so sorry, the units is incorporated

Comment: Line 91: p<0.000 Is it a mistake?

Response: its correct

Comment: Line 96 I propose to write down as Table 5a. instead of Table 5. (a):

Response: changed

Comment: Line 99: The same suggestion - Table 5b.

Response: changed

Line 111: Please insert the reference number instead of (EC, 2006).

Response: Changed

Comment: Line 141: 977+1000 ug/kg - Has standard deviation been calculated correctly?

Response: It is confirmed

Line 166: Please change it to Fusarium italica

Response: changed

Line 173-181: There are a few unit and editing errors i.e. µg/kg day b.w. instead of µg/kg b.w./day

Response: Thanks so much, the mistake is corrected

Line 193-194: Please correct this sentence

Response:  The sentence is changed as

“In-vivo study of DON levels in blood and urine in local consumers should be more desir-able for the assessment of exposure and implementation of regulations in population.”

Line 207: A pure standard of DON (100 mg/mL in ACN)… Please correct

Response: changed

Line 219: 5 minutes Please correct

Response: Corrected

Line 226: I suggest to (250 x 4.6 x 5 mm). Please correct 

Response: Changed

Line 236 and line 245: I suggest working on the patterns to improve their quality.

Response: The pattern is changed slightly

Line 267-268; 272; 283; 285; 292; 294; 298-299; 307 - I suggest to improve the literature items according to the journal's requirements

Response: The change is incorporated

Thanks once again for your time, critical comments and professional approach. We hope a positive outcome from here.

Round 2

Reviewer 1 Report

Page 1, line 17: Please delete: “A total of 2970 samples of twelve citrus fruits types were analyzed for patulin 17 (PAT) contamination”.

Page 1, line 42: Please clarify the samples. (…”high levels of DON were documented”… in corn grain samples).

Page 2, line 45-55: I’d include a comment about the corn products analyzed in this work (Iqbal et al, 2020).

Page 4, 92. What variety of corn do the authors refer to?  Also replace verity for variety.

Page 5, line 105. It would be interesting include the EU limits in the caption.

Figure 1: It seems that the figure 1 is duplicated.  In addition, the authors should be consistent with the name of the corn samples.

Page 9, line 157. Please check… “In our previous study [18], Iqbal et al. [18] we have investigated the seasonal variation of DON”…

Author Response

Please find enclosed our manuscript toxins-1111627 R2 titled “Variation of Deoxynivalenol Levels in Corn and Its Products Available in Retail Markets of Punjab, Pakistan and Estimation of Risk Assessment. The manuscript is revised again according to the comments raised by Reviewer no 1. We are really thankful for the comments/suggestions which not only increase the quality of our manuscript, but it also smoothens the flow and readability. Thanks

Reviewer 1

Comment: Page 1, line 17: Please delete: “A total of 2970 samples of twelve citrus fruits types were analyzed for patulin 17 (PAT) contamination”.

Response: So sorry, I remembered that I have deleted the line, also for other reviewer I have delete age word in Figure 1, but don’t know why these lines still appears. Anyway, I have deleted the mentioned line.

During revision, my method is first I do change that comment in manuscript and write its response as reviewer response section.

Comment: Page 1, line 42: Please clarify the samples. (…”high levels of DON were documented”… in corn grain samples).

Response: The sentence is changed as “ In previous study, 63 out of 100 samples were found to be contaminated with DON as documented by Raza et al. [17]”

Comment: Page 2, line 45-55: I’d include a comment about the corn products analyzed in this work (Iqbal et al, 2020).

Response: In this paragraph, the detail about wheat and wheat products samples were deleted and the sentence is changed as

“The results have shown that 87 (61.2%) samples of corn and corn products from summer and 57 (44.5%) from winter season were found to be contaminated with concentration ranged from 50 to 2967 µg/kg, and 50 to 2490 µg/kg, respectively.”

Page 4, 92. What variety of corn do the authors refer to?  Also replace verity for variety.

Response: sorry the mistake was corrected and changed as

“Corn 1= corn variety

Page 5, line 105. It would be interesting include the EU limits in the caption.

Response: The required limits are included, thanks

Figure 1: It seems that the figure 1 is duplicated.  In addition, the authors should be consistent with the name of the corn samples.

Comment: The figure 1 which appears duplicate has been deleted and corn products samples were arranged in same manner. Thanks for printout the mistakes.

Page 9, line 157. Please check… “In our previous study [18], Iqbal et al. [18] we have investigated the seasonal variation of DON”…

Response: Changed accordingly, Thanks once again for your great review and many sincere wishes from our side, stay healthy and active as always.

Reviewer 3 Report

Dear authors,

I thank you for your effort in drafting your answers. Unfortunately, I still see some flaws in the revised version:

- in Line 17 is still the sentence:  A total of 2970 samples of twelve citrus fruits types were analyzed for patulin 17 (PAT) contamination.

- All the sentences in which you delete the word about are now indistinguishable, e.g.
Line 19: Samples 539 (44.2%) of corn &and corn products were found to be positive with DON (n ≥ LOD).

- Yellow marked text, Line 52 in 54: use the number instead of LOD.

- Figure 1: y-axis - % age – you did not delete “age”.

- I cannot agree with the statement about the method of sampling. As you know, the method of sampling is very important because you need to achieve representativeness of the sample and further in the laboratory, homogeneity of the sample. What is a "A simple random methodology"? Did you take samples from different parts with special equipment, mix them and then prepare the final sample? Please describe this in more detail.

Author Response

Please find enclosed our manuscript toxins-1111627 R2 titled “Variation of Deoxynivalenol Levels in Corn and Its Products Available in Retail Markets of Punjab, Pakistan and Estimation of Risk Assessment. The manuscript is revised again according to the comments raised by Reviewer no 3. We are thankful for the comments/suggestions, which not only increase the quality of our manuscript, but it also smoothens the flow and readability of our manuscript. Thanks you so much.

Reviewer 3

Comments: I thank you for your effort in drafting your answers. Unfortunately, I still see some flaws in the revised version:

Response: So sorry and we have done our best in this revision that all suggested changes will be made according to your suggestions. Thanks for your time and efforts, once again.

- in Line 17 is still the sentence:  A total of 2970 samples of twelve citrus fruits types were analyzed for patulin 17 (PAT) contamination.

Response: I have deleted this line, I remembered. But, I have deleted again, sorry for inconvenience.

- All the sentences in which you delete the word about are now indistinguishable, e.g.
Line 19: Samples 539 (44.2%) of corn &and corn products were found to be positive with DON (n ≥ LOD).

Response: In second line, we have mentioned those samples which have levels higher than EU limits and changed as

“Samples 92 (7.5%) of corn and corn products having levels of DON greater than the proposed limits of EU (1000 µg/kg; 750 µg/kg)

- Yellow marked text, Line 52 in 54: use the number instead of LOD.

Response: Changed accordingly

- Figure 1: y-axis - % age – you did not delete “age”.

Response: I think, may be due to track changes, the figure 1 appears as duplicate. However, the mistake was removed.

- I cannot agree with the statement about the method of sampling. As you know, the method of sampling is very important because you need to achieve representativeness of the sample and further in the laboratory, homogeneity of the sample. What is a "A simple random methodology"? Did you take samples from different parts with special equipment, mix them and then prepare the final sample? Please describe this in more detail.

Response: We are extremely so sorry, in revised version, we have entered more detail about sampling procedure as “

A simple random methodology (each portion or lot has equal chance to be included) was used for collecting corn and corn products samples from farmers, market, and superstores. The gross samples were taken by hands and then homogenized and proper labelling was done. These three districts are most famous for producing corn in Punjab, Pakistan. Due to the high variability of fungi and mycotoxins in kernel samples, each sample size was not less than 2 kg. Then the kernel samples were mixed and ground in adequate particle size (Retsch, ZM 200, Haan, Düsseldorf, Germany). The lab sample portion was stored in plastic bags and kept in the freezer at -20 °C.

We are really thankful for reviewer no 3, for his time, efforts and professional responsibility and dedicated commitment. Thanks, and stay healthy and active as always.